# The Potts-Ising model for discrete multivariate data

**Zahra S. Razaee**
Biostatistics and Bioinformatics Research Center,
Cedars Sinai
zahra.razaee@cshs.org

**Arash A. Amini**
Department of Statistics
University of California, Los Angeles
aaamini@ucla.edu

## Abstract

Modeling dependencies in multivariate discrete data is a challenging problem, especially in high dimensions. The Potts model is a versatile such model, suitable when each coordinate is a categorical variable. However, the full Potts model has too many parameters to be accurately fit when the number of categories is large. We introduce a variation on the Potts model that allows for general categorical marginals and Ising-type multivariate dependence. This reduces the number of parameters from $\Omega(d^2 K^2)$ in the full Potts model to $O(d^2 + Kd)$, where $K$ is the number of categories and $d$ is the dimension of the data. We show that the complexity of fitting this new Potts-Ising model is the same as that of an Ising model. In particular, adopting the neighborhood regression framework, the model can be fit by solving $d$ separate logistic regressions. We demonstrate the ability of the model to capture multivariate dependencies in real data by comparing with existing approaches.

## 1 Introduction

Modeling multivariate discrete data is a basic problem in statistics and machine learning, especially in high dimensions. Discrete data are rarely independent and a fundamental modeling task is to characterize dependencies (correlation, causation, conditional independence, etc.) among variables. The problem is exacerbated when the data are high-dimensional. One of the most flexible tools available for modeling multivariate distributions are graphical models. The Potts model [1] is a versatile such model for discrete data, suitable when each coordinate is a categorical variable. The categorical nature of the model provides much more flexibility over Poisson-type count models. The Potts model is the natural extension of the well-known Ising model for binary data [2].

The full Potts model, however, has too many parameters to be accurately fit when the number of categories $K$ is large. The large number of parameters also precludes easy interpretation. We introduce a variation on the Potts model that allows for general categorical marginals and an Ising-type multivariate dependence. This reduces the number of parameters from $\Omega(d^2 K^2)$ in the full Potts model to $O(d^2 + Kd)$, where $d$ is the dimension of the data.

Our motivating example is the toxicity data collected in cancer clinical trials. Patients undergoing treatment may experience multiple toxicity types (such as nausea, diarrhea, etc) with grades of severity varying from 1, corresponding to a mild symptom, to 5 indicating death. The overall number of toxicities observed in cancer clinical trial is often more than 100. Furthermore, there is a rich dependence structure in toxicity data. For example, diarrhea can cause dehydration which leads to hypokalemia, hence the three toxicities are likely positively correlated, while some toxicities such as diarrhea and constipation are negatively correlated. The toxicity datasets are both high-dimensional and with rich dependencies.

The variation on the Potts model that we propose, which we refer to as the Potts-Ising model (POIS), has several attractive features. The marginals of the distribution are modeled after general

multinationals, hence provide a much better fit than Poisson, to sparse data with limited range. Moreover, the dependence structure is much simpler than the full model, captured by only a single matrix $\Gamma$ which allows for easy interpretation. The fitting of the model is no harder than fitting $d$ logistic regression problems. One can choose from a versatile array of regularization techniques to produce sparse or highly interpretable estimates of $\Gamma$, a surrogate for the correlation matrix.

Our model is suitable for any discrete data that has a special level, usually denoted as "0", and the dependence is captured by the presence of this level or its absence. The model is, for example, good for rating (or survey) data, where one level often has a special meaning. For example, in rating movies, products, etc., 0 often means "not rated" rather than the lowest rating (which starts at 1). Another example is the toxicity data, where "0" is significantly different from a rating of 1 or above, signifying that no symptom was observed for that particular adverse event. Although we take 0 to be the special level, any other level could work: our model is categorical, hence, any level can be designated as "0". Our model also work surprisingly well with count data as long as they are sparse enough so that the total number of unique values observed in the dataset is relatively low. We demonstrate all the above points in the sequel and show the effectiveness of the model with extensive simulations and comparison with a wide selection of existing approaches. The code for these simulations is available at GitHub repository `aaamini/pois_comparisons`.

**Related work.** The Ising model [3] for binary data and its extension, the Potts model, are well-known statistical models for discrete multivariate data [4, 5, 6]. Ravikumar et. al. [7] has shown that the $\ell_1$-penalized neighborhood regression in the Ising model can achieve high-dimensional model selection consistency, in analogy to the work of [8] for the Gaussian graphical models. Their algorithm was used in [9] to iteratively approximate the full likelihood by a series of pseudo-likelihoods estimated by neighborhood selection. In [10], an Ising model was used to detect the association between the US senators from their binary voting patterns. A sparse covariate-dependent Ising model was proposed in [11] to study conditional dependence within the binary data and its relationship with the additional covariates. The Potts model has been used in [2] to improve contact prediction between amino acids in protein chains. A quasi-Bayesian approach was proposed [12] to fit large Potts models with spike-and-slab priors to encode sparsity. The existing approaches either consider a Potts-type model with count variables or the full categorical Potts model. Our approach is categorical (hence more flexible than count modeling) but simpler than the full classical Potts model. Many other multivariate count models, based on extensions of the Poisson distribution, have been introduced in the literature. Inouye et. al. [13] provide a comprehensive review and comparison. We compare with a sizable selection of these approaches in Section 4 to which we defer the further discussion of these models.

## 2 The Potts-Ising model

Consider a discrete random vector $z = (z_i)_{i=1}^d$ taking values in $\mathbb{Z}_K^d := \{0, \ldots, K\}^d$, that is, each coordinate can take values $0, \ldots, K$. The general Potts model for $z$ assumes the following (joint) density,

$$p(z) \propto \exp\Big( \sum_{i=1}^d \sum_{k=0}^K \theta_{ik} z_{ik} + \sum_{\substack{i,j=1, \\ i<j}}^d \sum_{k,l=0}^K \gamma_{ij,kl} z_{ik} z_{jl} \Big), \quad z \in \mathbb{Z}_K^d \tag{1}$$

where $z_{ik} = 1\{z_i = k\}$. One could also impose a constraint on interaction parameters of the form $\gamma_{ij} = 0, (i, j) \notin E$, for some edge set $E \subset [d]^2$. Here, we do not assume such a priori constraint explicitly, although all the discussions easily extend to the case of a given edge set $E$. Model (1) is extremely flexible in capturing a multivariate dependence structure among the coordinates of $z$. This flexibility, however, comes at a cost: the number of parameters of the model is of the order of $d^2 K^2$, which is quite large if either the dimension $d$ or the number of levels $K$ is large. The drawbacks are two-fold; from a statistical perspective, the sample size needed to accurately estimate the model is high; from a computational perspective, fitting the model will be slow and prone to numerical instability for small sample sizes.

In this paper, we consider a restriction of (1) that preserves much of the flexibility of modeling pairwise interactions among the variables, but significantly reduces the model complexity. Our approach is suitable for sparse random vectors where level 0 has a special meaning. Instead of

modeling the interactions among all levels, we simply model the interaction between level 0 and not-0. To be more precise, let

$$\sigma_i := \sigma_i(z_i) := \begin{cases} 1 & z_i = 0 \\ -1 & z_i \neq 0 \end{cases}, \tag{2}$$

and consider the model

$$p(z) \propto \exp\Big( \sum_{i,k} \theta_{ik} z_{ik} + \sum_{(i,j):i<j} \gamma_{ij}\sigma_i\sigma_j \Big). \tag{3}$$

which we refer to as the Potts-Ising (POIS) model. The rationale behind the naming is that the interaction term in (3) is similar to an Ising model which is the special case of the Potts model for $K = 1$. We, however, note that (3) is much richer than an Ising model because it allows for modeling arbitrary marginal distributions for the coordinates via parameters $\theta_{ik}, i \in [d], k \in [K]$. We will show that we can achieve this extra flexibility over the Ising model, at virtually no extra computational cost. Model (3) is a special case of the general Potts model with the following restriction on the parameters:

$$\gamma_{ij,kl} := \begin{cases} \gamma_{ij} & k = l = 0, \text{ or } k \neq 0, l \neq 0 \\ -\gamma_{ij} & k = 0, l \neq 0 \text{ or } k \neq 0, l = 0 \end{cases}.$$

Through empirical validation on real data, we show that much of the statistical flexibility of the Potts model is also retained. In fact, the model in most cases achieves a performance similar to the ideal nonparametric benchmark as discussed in Section 4.

## 3 Model fitting

Consider a random sample $\{z^t\}_{t=1}^n$ of size $n$ where each $z^t$ is an i.i.d. draw with the same distribution as that of $z$ given in (1). We let $z_i^t$ be the $i$th coordinate of $z^t$ and write $z_i^* := (z_i^t, t \in [n])$. For the vector $z$, let $z_{-i} = (z_j, j \neq i)$ and similarly for $z_{-i}^* = (z_j^t, t \in [n], j \neq i)$ and $z_{-i}^t$. We gather the interaction parameters of (3) in the $d \times d$ matrix $\Gamma = (\gamma_{ij})$ and the marginal parameters in the $d \times K$ matrix $B = (\beta_{ik})$.

An effective approach for fitting graphical models of the form (1) is via the so-called *neighborhood regression* [8]: One separately fits the conditional distributions $p(z_i^* \mid z_{-i}^*)$ for all $i \in [d]$—here and in the sequel we use $p(x \mid y)$ to denote the conditional density of random vector $x$ given random vector $y$, also evaluated at realized values $x$ and $y$; this abuse of notation helps with brevity. Since these conditional densities factorize over the sample, i.e., $p(z_i^* \mid z_{-i}^*) = \prod_t p(z_i^t \mid z_{-i}^t)$, let us focus on the generic version based on $z$, i.e.,

$$p(z_i \mid z_{-i}) \propto \exp\Big( \sum_k \theta_{ik} z_{ik} + \sum_{j:\,j\neq i} \gamma_{ij}\sigma_i\sigma_j \Big).$$

Let us write $q_{ik} := \exp(\theta_{ik})$, $\gamma_{ii} = 0$, $\gamma_{i*} = (\gamma_{ij}, i = 1, \ldots, d)$ and $\sigma = (\sigma_j, j \in [d])$. Then,

$$p(z_i = k \mid z_{-i}) \propto \begin{cases} q_{i0}e^{\langle\gamma_{i*},\sigma\rangle}, & z_i = 0 \\ q_{ik}e^{-\langle\gamma_{i*},\sigma\rangle}, & z_i = k \neq 0 \end{cases}$$

where $\langle\gamma_{i*},\sigma\rangle = \sum_{j\neq i}\gamma_{ij}\sigma_j$ is the usual Euclidean inner product between $\gamma_{i*}$ and $\sigma$. Letting $\beta_{ik} := q_{ik}/q_{i0}$, and some algebra gives

$$p(z_i \mid z_{-i}) = \frac{\prod_{k\neq 0}[\beta_{ik}e^{-2\langle\gamma_{i*},\sigma\rangle}]^{z_{ik}}}{1 + \beta_{i\oplus}e^{-2\langle\gamma_{i*},\sigma\rangle}},$$

where $\beta_{i\oplus} := \sum_{k'\neq 0}\beta_{ik'}$. The notation $\oplus$ means that we are summing over all values of the index except 0. This notation is helpful because of the special role level 0 plays in the model. Later we use $+$ in place of an index to mean summing over all values of that index (without exception). Since $\beta_{i0} = 1$, in these notations, we have $\beta_{i\oplus} = \beta_{i+} - 1$.

### 3.1 Reduction to logistic regression

Let us go back to the sample of size $n$. With $z_{ik}^t = 1\{z_i^t = k\}$, we have

$$p(z_i^* \mid z_{-i}^*) := \prod_{t=1}^n p(z_i^t \mid z_{-i}^t) = \prod_{t=1}^n \frac{\prod_{k \neq 0}[\beta_{ik}e^{-2\langle\gamma_{i*},\sigma_t\rangle}]^{z_{ik}^t}}{1 + \beta_{i\oplus}e^{-2\langle\gamma_{i*},\sigma_t\rangle}}.$$

Here $\sigma_t = (\sigma_{tj}, j \in [n])$ where $\sigma_{tj} = \sigma(z_{tj})$ and $\sigma(\cdot)$ is defined in (2). The conditional log-likelihood of the model, $\ell_i(\beta_i, \gamma_{i*}) := \log p(z_i^* \mid z_{-i}^*)$, is

$$\ell_i(\beta_i, \gamma_{i*}) = \sum_{k \neq 0} z_{ik}^+ \log \beta_{ik} - \sum_t \left[2z_{i\oplus}^t\langle\gamma_{i*},\sigma_t\rangle + \log\left(1 + \beta_{i\oplus}e^{-2\langle\gamma_{i*},\sigma_t\rangle}\right)\right] \tag{4}$$

where $z_{i\oplus}^t = \sum_{k \neq 0} z_{ik}^t$ and $z_{ik}^+ = \sum_{t=1}^n z_{ik}^t$, using the summation convention discussed earlier.

To estimate the parameters, one often maximizes a penalized version of the the conditional log-likelihood. To keep the discussion simple, let us assume for now that there is no added penalty.

Given $\beta_{i\oplus}$, the problems of estimating $\beta_{i*}$ and $\gamma_{i*}$ decouple. A strategy is to solve the problem under the additional constraint $\beta_{i\oplus} = e^u$ for some $u \in \mathbb{R}$, and then optimize jointly over $u$ and $\gamma_{i*}$:

$$\max_{\beta_i > 0, \gamma_{i*}} \ell_i(\beta_i, \gamma_{i*}) = \max_{u, \gamma_{i*}} \max_{\beta_i > 0: \sum_{k \neq 0}\beta_{ik}=e^u} \ell_i(\beta_i, \gamma_{i*}).$$

The solution of the inner optimization problem over $\beta$ is simply

$$\widehat{\beta}_{ik}(u) = e^u \frac{z_{ik}^+}{z_{i\oplus}^+}, \quad k \neq 0, \tag{5}$$

where $z_{i\oplus}^+ = \sum_{k \neq 0} z_{ik}^+ = \sum_{t=1}^n \sum_{k \neq 0} z_{ik}^t = n - z_{i0}^+$, where the last equation follows since, by definition, $\sum_{k=0}^K z_{ik}^t = 1$ for all $t \in [n]$. Plugging-in, after some algebra, we have

$$\max_{\beta > 0, \gamma_{i*}} \ell_i(\beta, \gamma_{i*}) = C + \max_{u, \gamma_{i*}} \left\{ z_{i\oplus}^+ u - \sum_t \left[2z_{i\oplus}^t\langle\gamma_{i*},\sigma_t\rangle + \log(1 + e^{u-2\langle\gamma_{i*},\sigma_t\rangle})\right]\right\}$$

$$= C - \min_{u, \gamma_{i*}} \sum_t \left[z_{i\oplus}^t(2\langle\gamma_{i*},\sigma_t\rangle - u) + \log\left(1 + e^{u-2\langle\gamma_{i*},\sigma_t\rangle}\right)\right]$$

where $C = \sum_{k \neq 0} z_{ik}^+ \log(z_{ik}^+/z_{i\oplus}^+)$. The above is a convex problem jointly in $u$ and $\gamma_{i*}$.

To simplify, let $\widetilde{x} = (-2\gamma_{ij}, j \neq i)$, $\widetilde{a}_t = (\sigma_{tj}, j \neq i)$, and

$$x = \begin{pmatrix} u \\ \widetilde{x} \end{pmatrix}, \quad a_t = \begin{pmatrix} 1 \\ \widetilde{a}_t \end{pmatrix}, \quad b_t = z_{i\oplus}^t$$

so that $u - 2\langle\gamma_{i*},\sigma_t\rangle = \langle x, a_t\rangle$. Note that $b_t = 1 - z_{i0}^t$. The optimization can be written as

$$\min_{x \in \mathbb{R}^d} \sum_t \left[-b_t\langle x, a_t\rangle + \log\left(1 + e^{\langle x, a_t\rangle}\right)\right]$$

which is exactly the optimization problem for computing the MLE in a logistic regression model, based on the data $(b_t, a_t), t \in [n]$. Note that parameter $u$ in (5) is the intercept in this logistic regression problem. Once the logistic regression is fitted, we get the estimate $\widehat{\gamma}_{i*}$ of $\gamma_{i*}$ and $\widehat{u}$ of $u$. We can then use $\widehat{u}$ in (5) to obtain the estimates $\widehat{\beta}_{i*}(\widehat{u})$ of the marginal parameters. An alternative approach to optimizing the conditional log-likelihood, using the coordinate-descent, is described in the Supplement. We, however, found the above *global* approach to work better in practice.

Since all the parameters can be estimated by performing neighborhood regressions $p(z_i^* \mid z_{-i}^*)$ for all $i \in [d]$, the problem of fitting (3) reduces to performing $d$ parallel logistic regressions. The $i$th problem estimates the $i$th rows of $\Gamma$ and $B$. Since $\Gamma$ is symmetric, we obtain two estimates for each row/column. This is a common scenario in neighborhood regression and there are various rules available for combining these estimates. Here, we simply take the average.

**Regularization.** Due to the reduction of the (conditional) neighborhood regression to the logistic regression, any regularization technique available for the latter is immediately applicable to the former. In particular, we can easily minimize $(\beta_i, \gamma_{i*}) \mapsto -\log \ell_i(\beta_i, \gamma_{i*}) + \rho_{\lambda_i}(\gamma_{i*})$ for a penalty $\rho_\lambda$ where $\lambda$ is a regularization parameter. Following through with the argument of Section 3.1, this just adds $\rho_{\lambda_i}(\gamma_{i*})$ to the logistic regression objective. One can also think of adding the penalty as putting a prior, namely, one that is $\propto e^{-\rho_\lambda(\cdot)}$, on $\gamma_{i*}$, in a Bayesian interpretation of the problem.

We consider three choices: (1) The sparsity-inducing $\ell_1$ penalty, $\rho_\lambda(\cdot) = \lambda \| \cdot \|_1$, (2) Firth's biased-reducing penalty $\rho_\lambda(\cdot) = -\frac{1}{2} \log \det I(\cdot)$, where $I(\cdot)$ is the Fisher information of the logistic problem [14]; and (3) the default prior of the `arm` R package [15]. Firth's approach is equivalent to putting a Jeffrey's prior on $\gamma_{i*}$ and we use the implementation available in `logistf` R package [16]. The three approaches are implemented in the accompanying code. In the paper, we focus on the the the $\ell_1$ penalty as the default choice, due to its robustness, and interpretability in high dimensions, as well as the availability of fast implementations such as the `glmnet` R package [17]; we take advantage of the efficiency of `glmnet` in using warm starts to compute the entire regularization path as of function of $\lambda$. We can then use cross-validation to tune $\lambda$ as will be discussed in Section 4.

# 4 Empirical results

We now present results on the performance of the model on real data. We adopt the framework of Inouye et. al. [13] who have done extensive simulations comparing various methods on discrete multivariate data. Among the methods they considered, we compare with the Copula Poisson (COPPOI)—estimated via the two-stage IFM method [18] via the DT [19], (MIXPOI), the Truncated Poisson Graphical Model (T-PGM) [20], independently-fit Poissons (INDPOI), a log-normal model (LOGNORM) and independently-fit negative binomials (INDNEGBIN). For the implementation of these methods, we have relied on the open-source code provided by Inouye et. al. in [21].

We also introduce and compare with four new methods: the Bootstrap, the copula multinomial (COPMULT), the independently-fit multinomials (INDMULT) and the global POIS solution (POIS)—as discussed in Section 3.1—with the $\ell_1$ regularization implemented via `glmnet` package. The details of the Bootstrap are discussed below. We also did experiments with the Poisson Square Root (POISQR) model [22] and a log-normal model (LOGNORM). The POISQR was a hit and miss (with more misses in the datasets that we considered) while the LOGNORM performed poorly across the board. We have excluded these two methods from the results due to their high computational complexity.

We consider the following publicly available datasets: MovieLens 100K Dataset [23] of movie ratings ($n = 1037$, sparsity $\approx 97.8\%$), and an Amazon customer ratings dataset [24] ($n = 745$, sparsity $\approx 94.3\%$). We also use datasets on cancer drug toxicities which provided the motivating examples for this work. The selected number of columns, $d$, is shown on the figures.

**Toxicity and questionnaire data.** We use the toxicity data from the NSABP R-04 colorectal cancer clinical trial [25, 26]. NSABP R-04 was a phase III trial (NCT00058474). The trial included 1,608 participants, with complete toxicity data available for $n = 1596$ patients. The toxicities are graded from 0 to 5, representing "no symptoms" to "death". These data are provided by our collaborators and not yet publicly available; see the acknowledgment section for more details. Patients were assigned to four different treatments, but here we look at the aggregate data across all treatments. We reduce the data to the most frequent $d = 36$ symptoms. The result is a $1596 \times 36$ data matrix, with entries in $\{0, 1 \ldots, 5\}$, and sparsity of $\approx 94\%$. This type of data is very suitable for the application of the POIS model due to its limited range and sparsity.

Quality-of-life questionnaires, also known as Patient Reported Outcome (PRO) questionnaires, are another commonly collected data in cancer trials. PRO data were collected for patients enrolled in NSABP R-04 colorectal cancer clinical trial prior to treatment, at the end of chemoradiation prior to surgery, and then 12 months after surgery [25]. The patients answered the questions with: Not at all, A little bit, Somewhat, Quite a bit, Very much, that are mapped to 0 to 4. Here, we look at the aggregate PRO data, across all time points, resulting in a data matrix of size $3295 \times 17$, after restricting to the most frequent symptoms. The sparsity of PRO data is $\approx 77\%$. We will also use the toxicity and PRO data for the breast cancer from the same study, the details of which are presented in the Supplement.

**Evaluation criteria.** We evaluate the fit of the models using the maximum mean discrepancy (MMD) [27], which measures the maximum difference between general moments of two distributions. More specifically, for two probability measures $\mathbb{Q}$ and $\mathbb{P}$, the MMD is defined as

$$\delta(\mathbb{Q},\mathbb{P}) = \sup_{g \in \mathcal{G}} |\mathbb{E}g(X) - \mathbb{E}g(Y)|, \quad X \sim \mathbb{Q}, \ Y \sim \mathbb{P}$$

where $\mathcal{G}$ is a class of functions. Usually one takes $\mathcal{G}$ to be the unit ball of a universal reproducing Hilbert space (RKHS), in which case $\delta$ is a proper metric on the space of probability measures [27]. The MMD defined above is a single number. To get some robustness in empirical applications, we consider the MMD between all the $(d-2)$-dimensional marginals of $\mathbb{P}$ and $\mathbb{Q}$, assuming that the two distributions are $d$-dimensional. More precisely, letting $X = (X_1, X_2, \ldots, X_d)$, and similarly for $Y$, we compute

$$\Delta_{\mathcal{I}}(\mathbb{Q},\mathbb{P}) = \sup_{f \in \mathcal{F}} \left| \mathbb{E}f(X_{i_1}, X_{i_2}, \ldots, X_{i_{d-2}}) - \mathbb{E}f(Y_{i_1}, Y_{i_2}, \ldots, Y_{i_{d-2}}) \right|, \tag{6}$$

for all $\mathcal{I} := \{i_1, \ldots, i_{d-1}\} \subset \{1, \ldots, d\}$. Here, $\mathcal{F}$ is the unit ball of an RKHS of functions on $\mathbb{R}^{d-2}$, which in this paper will be taken to be the RKHS of a Gaussian kernel. In simulations, we report the histogram of $\Delta_{\mathcal{I}}$ as $\mathcal{I}$ varies over all possible $\binom{d}{d-2}$ choices. Our approach here resembles that of Inouye et. al. [22], with the key difference that we consider $(d-2)$-dimensional marginals, whereas they consider the 2-dimensional ones, that is, pairwise moments of the form $\mathbb{E}f(X_{i_1}, X_{i_2})$. In practice, pairwise dependence might not be strong for many pairs, but the overall distribution could be far from a product distribution due to higher-order dependencies. Our approach has the advantage of measuring higher-dimensional dependencies while retaining the aggregation idea introduced by Inouye et. al. If the RKHS is rich enough, functions of the form $f(X_{i_1}, X_{i_2}, \ldots, X_{i_{d-2}})$ already include those of the form $f(X_{i_1}, X_{i_2})$ by being constant in the extra arguments, so the pairwise dependence is also implicitly measured by (6).

In the simulations, we split the data into a training and a test set. We take $\mathbb{P} = \mathbb{Q}^{\text{test}}$, the empirical distribution of the test set, as a surrogate for the true data-generating distribution $\mathbb{Q}^*$. A parametric model $\mathbb{Q} = \mathbb{Q}_\theta$ will depend on some parameter $\theta$ which is estimated based on the training set to give us $\widehat{\theta}$. Ideally, we would like to evaluate $\Delta_{\mathcal{I}}(\mathbb{Q}_{\widehat{\theta}}, \mathbb{Q}^{\text{test}})$. However, since the exact computation is often intractable, we generate a sample of size $m$ form $\mathbb{Q}_{\widehat{\theta}}$, and use its empirical distribution $\hat{\mathbb{Q}}_{\widehat{\theta}}$ in place of $\mathbb{Q}_{\widehat{\theta}}$. The histogram plots we show are that of $\Delta_{\mathcal{I}}(\hat{\mathbb{Q}}_{\widehat{\theta}}, \mathbb{Q}^{\text{test}})$.

We also consider a nonparametric approach, the Bootstrap, that models the distribution of the training set by its empirical distribution $\mathbb{Q}^{\text{train}}$. The performance in this case is ideally measured by $\Delta_{\mathcal{I}}(\mathbb{Q}^{\text{train}}, \mathbb{Q}^{\text{test}})$. To be consistent with the calculations in the parametric case, we instead generate a sample of size $m$ from $\mathbb{Q}^{\text{train}}$ and form its empirical distribution, denoted as $\hat{\mathbb{Q}}^{\text{train}}$. This amounts to a resampling of the original (training) data, hence the name bootstrap. We then evaluate the histogram of $\Delta_{\mathcal{I}}(\hat{\mathbb{Q}}^{\text{train}}, \mathbb{Q}^{\text{test}})$, which will serve as the benchmark for the best performance achievable (in terms of the MMD). The sampling parameter is generally taken to be $m = 1000$ in our simulations. For each pair-complement, we compute the MMD over a collection of Gaussian kernels with bandwidths varying in the range $10^{-2}$ to $10^{0.8}$ (15 points equally spaced on the log-scale) and take their mean. The results are further averaged over $n_{\text{CV}} = 5$ random training-test splits. The MMD calculations are based on the fast approximation using 64 random Fourier features [28, 29].

**Tuning.** The three methods POIS, T-PGM, and MıxPoı have hyper-parameters that we tuned by splitting the training set further into training and validation sets (at $70/30$ ratio) and evaluating the performance on the validation set using the pair-complement MMD. For the POIS model, we used a single regularization parameter $\lambda$ for the $\ell_1$ penalty in all the neighborhood regressions. We used 15 values of $\lambda$ between $10^{-4}$ to $10^{-1.3}$, equally spaced on the log-scale. For each $\lambda$, we estimated both the $\hat{\Theta}(\lambda)$ and $\hat{\Gamma}(\lambda)$ matrices. Then, to determine the optimal $\lambda$, we sampled from each model $(\hat{\Theta}(\lambda), \hat{\Gamma}(\lambda))$ and compared the samples to the validation set using the pair-complement MMD. An alternative approach would have been to use and tune a different $\lambda$ for each neighborhood regression, using the standard cross-validation for regression. We found that the former approach gives more accurate results and suffers less from the $\ell_1$ shrinkage issue. For the MıxPoı, the tuning parameter is the number of mixture components, for which we used a range of five values between 5 to 30, equally spaced on the log-scale. For T-PGM, we used the same tuning code as Inouye et. al. [13].

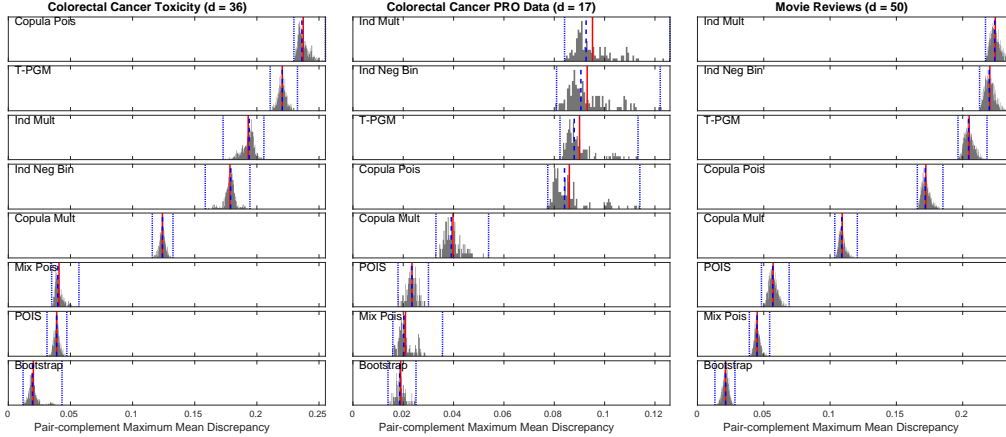

Figure 1: Pair-complement MMD histograms for three datasets.

## 4.1 Results

Figure 1shows the pair-complement MMD histograms for various methods on the movie ratings, and the colorectal cancer toxicity and PRO data. Similar plots for the breast cancer and Amazon data are presented in the Supplement. Here, we have used the `glment` regularization for the POIS model. Among the methods, the POIS, Bootstrap and MIXPOI are the top three most well-fitted models in the majority of cases. Contrasting to the poor fit of the Poisson models (except for MIXPOI), the results suggest that multinomial models are much better suited for modeling sparse discrete data with a relatively limited range. That the Bootstrap often fits the best was expected, since it is a nonparametric approach (and the best model of the training data in the absence of any prior assumption); however, it provides no model for the data, hence no interpretation, and we mainly use it as a benchmark. MIXPOI fits very well in most cases too, but achieves this performance with many components, generally in the range of 20 to 30 for most cases, hence comparable to the Bootstrap in its lack of interpretation. The POIS model fits very well to these data, as the plots show. The upside is that the POIS model is highly interpretable as discussed below. We also note that POIS significantly outperforms the Copula Multinomial model on toxicity and PRO data.

In all cases, POIS uncovers interesting dependency structures in the data. Figure 2 shows the estimated $\Gamma$ matrix obtained by the POIS model on the colorectal cancer toxicity dataset. The corresponding estimated Spearman correlation matrix is illustrated in Figure 3 along with the Spearman correlation matrix computed from the original data. The estimated correlation matrix is constructed based on $2 \cdot 10^4$ samples generated from the estimated POIS model. There is a close agreement between the estimated and the original correlation matrices. An interesting observation is that the majority of the entries of these correlation matrices are nonnegative, with few entries having very small negative values. This suggests that the correlation matrix is not suited for uncovering reverse dependencies in this case. The estimated $\Gamma$ matrix however shows plenty of strong reverse dependencies via its negative entries. This is because $\Gamma = (\gamma_{ij})$ encodes different information from a correlation matrix, namely, it encodes pairwise conditional dependence (or partial correlation) between nodes. In other words, $\gamma_{ij}$ is related to the partial correlation of nodes $z_i$ and $z_j$ conditioned on $z_k, k \in \{i, j\}$. Based on Figure 3, this partial correlation is strongly negative for some pairs in the data.

Qualitatively, the $\Gamma$ matrix estimated by POIS makes a lot of sense. For example, some of the observed highly positive (conditional) associations, such as those between nausea (Naus) and vomiting (Vmtn), anal mucositis (Anlm) and rectal mucositis (Rctm), white blood cell decreased (Wbcd) and neutrophil count decreased (Ntcd), and hypoalbuminemia (Hypl) and hypocalcemia (Hypc) are all as expected. The negative (conditional) associations uncovered are quite interesting. For example, Figure 2 shows a strong negative association between hypokalemia (Hypk), that is, decreased Potassium concentration, and constipation (Cnst). Checking the data, we observed that there is a perfect dichotomy between these two symptoms: Of all the patients showing either of these two at various degrees, none shows both at the same time. A possible explanation is that Hypk is usually caused by dehydration (Dhyd) which can be caused by diarrhea (Drrh), the opposite of Cnst. (It is worth noting that severe Hypk can

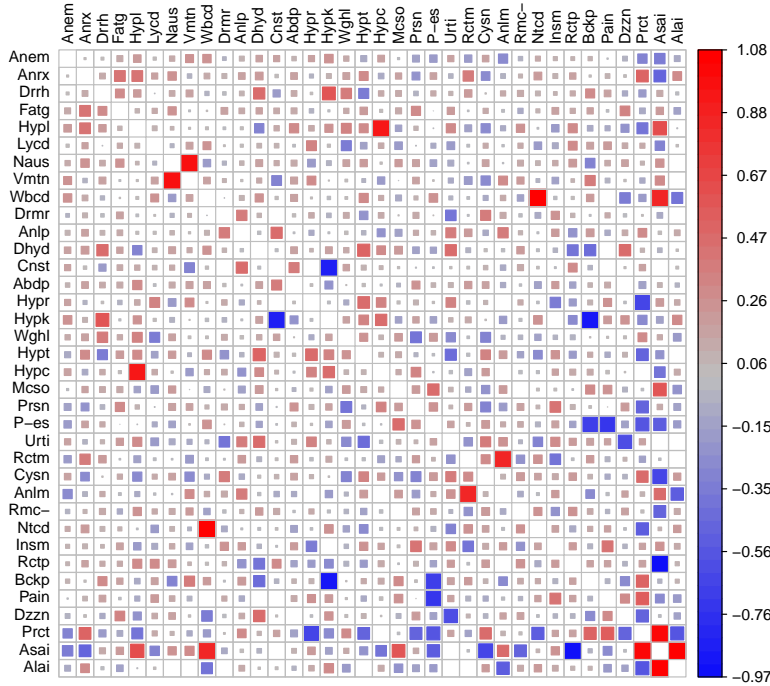

Figure 2: The estimated $\Gamma$ matrix for the colorectal cancer toxicity data. See Table 1 (in the Supplement) for the meaning of the abbreviations.

also cause Cnst, but this situation does not seem to have occurred in this dataset.) This reasoning also explains the high positive association between Hypk and Drrh. The other strong negative associations can also be traced back to near dichotomies in the dataset.

Figure 4 illustrates the runtime of the POIS algorithm versus the dimension $d$ and the sample size $n$. Since we are solving $d$ regression problems of size $n \times d$, we conjecture the complexity to be at most $O(nd^2)$, assuming $d \leq n$. This is corroborated by the plots in Figure 4 which in fact show a somewhat better average complexity of roughly $n^{0.55}d^{1.9}$.

**Discussion.** We proposed a variant of the Potts model that is more interpretable, has fewer parameters and can be easily fit using penalized logistic regression. We mainly considered the $\ell_1$-penalization which has the sparsity-inducing property desirable in high-dimensional settings. The $\ell_1$ penalty, however, is known to cause the shrinkage of the estimated parameters, and one has to be careful not to over-regularize. In practice, one can use values of $\lambda$ slightly lower than what is suggested by cross-validation, or use other known debiasing techniques, including refitting a low-dimensional model, without penalty, to the support uncovered by the $\ell_1$-penalized solution [30]. It is possible to generalize the POIS model to allow variable thresholds and arbitrary threshold functions without much difficulty; for example, we can take $\sigma_i(z_i)$ in (2) to $= 1\{z_i \leq \text{Median}(\{z_{it}\}_{t=1}^n)\}$, where $\text{Median}(\{z_{it}\}_{t=1}^n)$ is the empirical median of the $i$th column of the data, to bisect the range of each variable. For count data, or other discrete data with a highly variable range, we can consider a further discretization (i.e., binning) to make them suitable for our categorical model.

## 5    Broader Impact

In cancer clinical trials, patients are assigned to different treatment groups, and for each patient, toxicities are collected. These toxicities are graded, high-dimensional and correlated. Patient reported outcome questionnaires also collect patients' responses to quality of life questions on a Likert-type

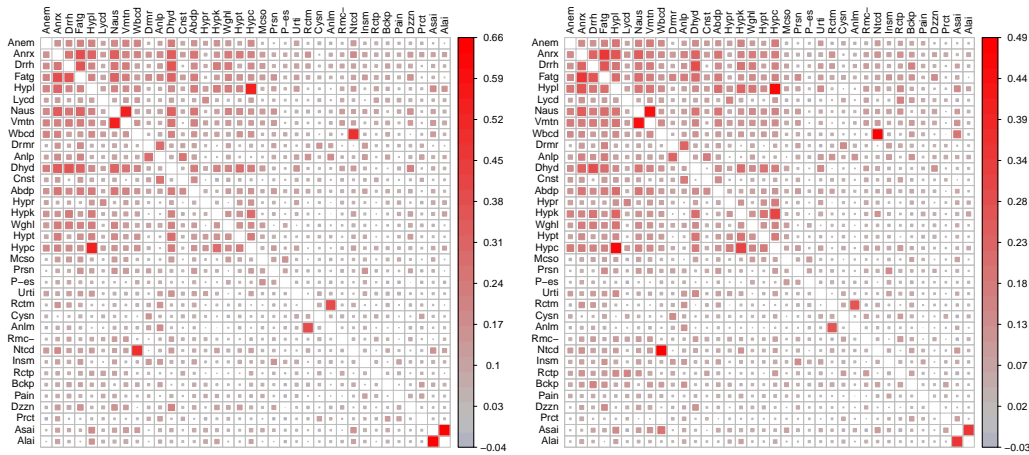

Figure 3: The Spearman correlation matrix for the original data (left) versus that of the estimated model (right). The estimated correlations were calculated based on a sample of size $2 \cdot 10^4$ from the fitted model.

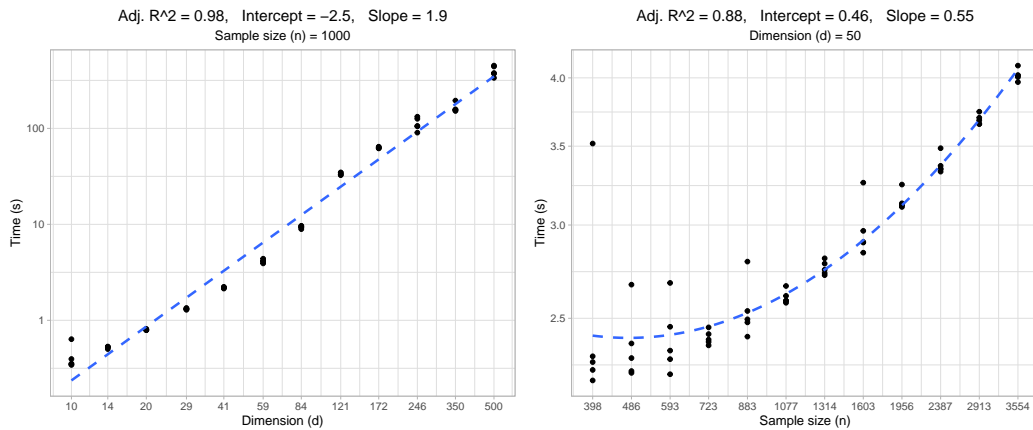

Figure 4: POIS runtime versus $d$ for fixed $n$ (left) and vice versa (right).

scale after treatments. It is crucial to correctly model these kind of data and estimate the main effects as well as the association between the toxicities, in order to determine the tolerability of treatments and their impact on patients quality of life. Our Potts-Ising model is a suitable such model designed for the toxicity data, but applicable far beyond it to any survey and rating data with limited range, as well as, sparse count data.

## Acknowledgments and Disclosure of Funding

This work was supported in part by the National Cancer Institute of the National Institutes of Health grant 1U01CA232859-01. Arash Amini was supported by the NSF CAREER grant DMS-1945667. The work was completed while the first author was a post-doc at the Cedars Sinai Medical Center under the supervision of André Rogatko whom we thank for the motivation behind this work and numerous helpful discussions. We also thank Mourad Tighiouart and Marcio Diniz for their support and Ron D. Hays for his comments. We would like to also thank Patricia A. Ganz, Ron D. Hays and Greg Yothers for sharing the toxicity and PRO data with us. We thank Inouye et. al. [13] for publicly sharing their code which allowed us to build on their work. Finally, we thank the anonymous reviewers for their constructive comments.

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
