[Supplementary Material]

# The Potts-Ising model for discrete multivariate data: Supplementary material

Zahra S. Razaee and Arash A. Amini

## 6  1.5-step block coordinate descent

An alternative approach to optimizing the conditional log-likelihood (4) is to alternate between solving for $\beta_{i\oplus}$ and solving for $\gamma_{i*}$, that is, performing a form of block coordinate descent. Initializing $\gamma_{i*} = 0$ and optimizing over $(\beta_{ik})_{k \neq 0}$, we obtain the closed form solution

$$\beta_{ik}^0 = z_{ik}^+/z_{i0}^+, \ k \neq 0.$$

We then fix $\beta_{ik} = \beta_{ik}^0$ and optimize $\ell_i(\beta^0; \gamma_{i*})$ over $\gamma_{i*}$. Let $u_0 = \log(\beta_{i\oplus}^0)$ where $\beta_{i\oplus}^0 = \sum_{k \neq 0} \beta_{ik}^0 = (n/z_{+i0}) - 1$. Then, the problem is equivalent to solving

$$\gamma_{i*}^0 = \underset{\gamma_{i*}}{\operatorname{argmin}} \sum_t \left[ z_{i\oplus}^t (2\langle \gamma_{i*}, \sigma_t \rangle - u_0) + \log(1 + e^{u_0 - 2\langle \gamma_{i*}, \sigma_t \rangle}) \right]$$

which is that of a logistic regression, with *fixed intercept* $u_0$. After obtaining $\gamma_{i*}^0$, we can maximize $\ell_i(\beta; \gamma_{i*}^0)$ over $\beta$, whose solution can be written as $\beta_{ik}^1 = z_{+ik}/x$ where $x$ solves the nonlinear equation:

$$\sum_{t=1}^n \frac{e^{-2\langle \gamma_{i*}^0, \sigma_t \rangle}}{x + e^{-2\langle \gamma_{i*}^0, \sigma_t \rangle} z_{i\oplus}^+} = 1.$$

This equation can be solved efficiently by bisection. One can then repeat the iterations. However, we found in practice that terminating after obtaining $(\beta_{ik}^1, \gamma_{i*}^0)$ is good enough. In fact, this early termination seems to have an implicit regularization effect.

## 7  Empirical results continued

We have also applied the methods to the breast cancer toxicity and PRO data from the same clinical cancer trial discussed in the text. This toxicity data has dimensions $3070 \times 45$ with sparsity $\approx 95\%$ and its corresponding data is $9079 \times 29$ with sparsity $56\%$. Figure 5 shows the results. POIS outperforms all other methods except the Bootstrap on the breast cancer toxicity data. The corresponding PRO data is the only dataset on which POIS is slightly less competitive relative to say Copula Multinomial. This can be explained by noting that the breast cancer PRO data is quite dense (56% sparsity), making the POIS model less suitable due to the violation of its underlying sparsity assumption.

Figure 6 shows the results on the Amazon rating data ($n_{\mathrm{CV}} = 2$) along with a larger plot of the Movie rating results presented in the text. Similar conclusions about the relative performance of the POIS model can be made as those discussed in the text. For completeness, Figure 7 provides larger plots for the colorectal cancer results, already presented in Figure 1 of the main text. Table 1 shows the abbreviations used for toxicities in Figure 2 of the text.

Figure 5: Pair-complement MMD histograms: Toxicity and PRO data (Breast Cancer).

Figure 6: Pair-complement MMD histograms: Ratings data.

| | | | |
|---|---|---|---|
| Anem | Anemia | Hypc | Hypocalcemia |
| Anrx | Anorexia | Mcso | Mucositis oral |
| Drrh | Diarrhea | Prsn | Peripheral sensory neuropathy |
| Fatg | Fatigue | P-es | Palmar-plantar erythrodysesthesia syndrome |
| Hypl | Hypoalbuminemia | Urti | Urinary tract infection |
| Lycd | Lymphocyte count decreased | Rctm | Rectal mucositis |
| Naus | Nausea | Cysn | Cystitis noninfective |
| Vmtn | Vomiting | Anlm | Anal mucositis |
| Wbcd | White blood cell decreased | Rmc- | Rash maculo-papular |
| Drmr | Dermatitis radiation | Ntcd | Neutrophil count decreased |
| Anlp | Anal pain | Insm | Insomnia |
| Dhyd | Dehydration | Rctp | Rectal pain |
| Cnst | Constipation | Bckp | Back pain |
| Abdp | Abdominal pain | Pain | Pain |
| Hypr | Hyperglycemia | Dzzn | Dizziness |
| Hypk | Hypokalemia | Prct | Proctitis |
| Wghl | Weight loss | Asai | Aspartate aminotransferase increased |
| Hypt | Hypotension | Alai | Alanine aminotransferase increased |

Table 1: Abbreviations used for the toxicity data.

Figure 7: Pair-complement MMD histograms: Toxicity and PRO data (Colorectal Cancer).