[Reviews · NeurIPS 2020]

Review 1

Summary and Contributions: This paper proposes a new model for multivariate data which aims at reducing the complexity of Potts Model for multivariate data. The core idea in the paper is that certain labels are special in multivariate data and the pairwise relationship between variables just depends on the fact whether it has a special label or non-special label and not on which particular non-special label is it.So, the complexity of pairwise term is reduced to an Ising model of binary data. The model still differs from Ising model in the sense that unary potentials are still different for each of the multivariate labels. So, the model complexity and capacity is higher than merging all non-special labels into a single label. The authors also show that MLE in this setting reduces to evaluating multiple logistic regression functions. They show that learning in this setting is highly effective making the learning problem highly efficient and show empirical gains by the approach. The main contribution of the paper is proposal of a new model and deriving algorithms for learning these by both block coordinate descent and reductions to multiple logistic regression problems.

Strengths: The work seems to catch an interesting insight in real world problems that certain labels are special and the complexity of pairwise multivariate models can be greatly reduced by utilizing this special property. They propose a new model called Potts Ising Model using this property. The examples and use cases for this newly proposed model seems convincing and useful for applicability in the real world for many domains. The model motivation, description and learning details are specified at appropriate detail making it easy to read till that section. I particularly like the motivation of the work based on cancer data and that the problem is motivated from a very real world context. The experiments on a real world dataset is a great plus which is missing in many of the current works in the Graphical Models community.

Weaknesses: I believe the experiments are rather weak and I would have liked to see some inference results rather than measuring only MMD on these datasets. I am not sure about the experimental methodology of comparing at distribution level without going to end inference results. I am not very clear on experimental details. I am assuming the pairwise connections in movie ratings data is considered between variables which have the same movie and variables which have the same user. Is it true or it is considered between all variables ? Also, how d=50 is chosen . Similar holds true for Book ratings dataset. A great utility of these models occur when the number of variables in the models is increased and there is repetitive structure. I understand the limitation on real world datasets of toxicity but No experiments beyond dimension 50 are provided even for movielens and other datasets. It would be great to have some results and discussion on how these models scale with increasing dimension. Post Rebuttal: Thanks for the detailed response to all the reviewers questions. I appreciate your effort on additional experiments and the detailed review. Overall, i liked the approach being simple and motivated from a real world context. I am happy to increase my score based on response and explanation but would definitely appreciate more experiments with larger number of variables.

Correctness: The methods and claims look correct. I am not sure if this is the right empirical methodology to evaluate such results. I would have liked to see some inference results from the learned models.

Clarity: The paper is well motivated but the mathematical notation is confusing in section 3. I would strongly suggest authors to avoid 3 $z_{itk}$ subscript indices. While one is able to figure out the notation when one spends time, I believe it can be highly simplified. I will suggest the authors to refer to the section of potts and ising model in [Koller and Friedman 2009] book for simplifying the notation and increasing accessibility for the reader. The experiments section have some missing details as specified above

Relation to Prior Work: Yes, I believe the previous work has been discussed in decent detail wrt the problem..

Reproducibility: Yes

Additional Feedback: I still believe some details are missing for experiments. Some of those details are specified above. Also, the paper says “The sampling parameter is generally taken to be m=1000 in our simulations”. In case of toxicity data, how it is 1000 when total n is of the order of 300. Am i misunderstanding something here ? These experimental details should be clearly specified.


Review 2

Summary and Contributions: This paper proposes a new model for sparse discrete data by modeling the independent parameters as categorical while simplifying the pairwise parameters to binary variables. The paper shows that both the categorical parameters and the pairwise parameters can be estimated via nodewise logistic regressions or alternating minimization. Finally, the paper presents empirical results on several datasets and compares to previous methods for count data.

Strengths: - Develops two algorithms for optimizing the proposed model. - Compares to multiple previous methods for discrete data (particularly count-based data). Many good baselines included.

Weaknesses: I appreciated the author response with the new experiments and new evaluation method. The new evaluation idea is very nice and does show a more complete picture. Also, I greatly appreciated the comparison to Copula Multinomial. In terms of computation time, CopMult is much faster (almost trivial) so this should be added to the discussion section but it seems that POIS-g does in fact do better in some circumstances. Also, the interpretability results could still use improvement and validation. I've updated my score based on this. ------ Original review - From the evaluations, it cannot consistently beat the Ind Mult baseline in terms of MMD. MMD is just as good for Independent Multinomial compared to the POIS approaches. - Lacks comparison to an important simple baseline: Copula Multinomial (i.e., use multinomial for marginals and Gaussian copula for copula using simple IFM method similar to CopPoi but with multinomial). This would likely be a very difficult baseline to beat as it would perform strictly better than Ind Multinomial. You could also use graphical lasso or the non-paranormal skeptic to estimate the Gaussian copula [1] in high-dimensional regime---and these are known to have very good theoretical rates. Additionally, this could also provide a copula correlation matrix or sparse inverse correlation matrix (similiar to the $\Lambda$ matrix of POIS). Finally, this approach would be super fast as it is probably no more than Copula Poisson + Ind Mult computation times. - Unclear if learned correlations/dependencies are intuitive/good or just spurious. More quantitative or qualitative justification via interpretation would improve the paper. See correctness below for more details. [1] Liu, Han; Han, Fang; Yuan, Ming; Lafferty, John; Wasserman, Larry. High-dimensional semiparametric Gaussian copula graphical models. Ann. Statist. 40 (2012), no. 4, 2293--2326.

Correctness: A more thorough understanding of the correlations is still needed. ------ Original review The evaluation method via MMD seems reasonable. The derivations seem correct. The paper claims that "POIS uncovers interesting correlation structures in the data". This is not substantiated in any qualitative or quantitative way except by showing the dependency matrices and saying that it has richer structure than spearman correlation but was not interpreted or explained further. These matrices may merely be spurious dependencies. Also, the correct comparison would probably be to the *inverse* of the spearman correlation matrix (which would correspond to the graphical model structure of a Gaussian copula model). Currently, the comparison is probably unreasonable (it's like comparing a matrix to its inverse). I would also want to see how this compares to graphical model structure of the Multinomial Copula model above.

Clarity: The technical part is very difficult to read especially with the many non-standard notations that are introduced. It is almost impossible to keep track of all the non-standard symbols and notation on a first read of the paper. The paper could be improved by simplifying notation. For example, it may be simpler to actually just use a summation instead of the circle plus symbol.

Relation to Prior Work: The relation to prior work is reasonable. However, the paper compares primarily to Poisson-based prior models. It is somewhat unsurprising that Poisson models do not work for these high sparsity datasets. Also, the paper uses review data that is on a scale of 5, which is bounded and thus inherently not like Poisson or negative binomial models. Even a binomial distribution with N=5 may be closer to the true distribution.

Reproducibility: Yes

Additional Feedback: Maybe use $\tilde{z}$ instead of $\sigma$ to help the reader remember that it is related to $z$. I'd suggest writing the idea of solving using logistic regression as a lemma or proposition. Then, describe the intuitive steps of the proof but put the actual proof in the appendix. The complex details of the proof make the paper hard to read and the derivation itself does not seem particularly insightful. Why is coordinate descent better than conditional log-likelihood? Is this just because you are jointly optimizing all the parameters instead of splitting them? An idea for model simplification: Could you write your model more simply using indicator functions? For example, you could write I(z_i = j) for the independent sufficient statistics and I(z_i \neq 0) I(z_j \neq 0) as the pairwise sufficient statistics. Typo?: Lines 204: Should the reference be [12] rather than [19]?


Review 3

Summary and Contributions: The authors discuss a special case of the Potts model that can be fit using the neighborhood regression framework in much the same way as Ising models (and with similar complexity).

Strengths: - This particular restriction is novel to me and the argument for why this model might make sense in practice is reasonable.

Weaknesses: - The restriction seems quite severe. One wonders if there are even slightly more complicated models that could improve over this result. In particular, the observation that this can be fit almost exactly like an Ising model isn't really surprising. - The analysis looks at a single data set. More detailed experiments are needed to draw the kinds of conclusions that are suggested by this work. This is particularly important for me as the motivation is that lots of applications have the feature necessary for this model to make sense. - I like the motivating problem, but it seems that you might want some correlations between the different classes (even if mild) on any real data. - The technical contributions quite closely follow existing work which limits their novelty. ---Post Rebuttal--- I'm still a bit skeptical that such a coarse approach would really work on a broad range of data sets - though there is enough variety to suggest that perhaps it does. The comment in the rebuttal, "It may not be as severe as it may seem," does little though to help me understand the counterintuitive (at least in my mind) nature of the experimental results. Does changing which level is designated as '0' impact the quality of the results?

Correctness: As far as I can tell.

Clarity: The paper is clear and easy to follow except for a few minor typos here and there.

Relation to Prior Work: There are lots of different special cases of the Potts model considered in the physics literature and beyond. It might be worth citing a few of these just for context.

Reproducibility: Yes

Additional Feedback:


Review 4

Summary and Contributions: This paper presents a new model, the Potts-Ising Model (POIS) to describe survery, rating, and sparse count data. The authors also provide a corresponding algorithm to learn the model from data. The work is highly motivated by a specific example from cancer drug clinical trials.

Strengths: - I find it very interesting and refreshing that the motivating example---toxicity data in cancer clinical trials---is discussed right away. This provides very good motivation for the work and clear set up of broader impacts. - The model is simple to understand but flexible, and fills a gap between simple Ising models and much more general Potts models. - Testing on 4 publicly available data sets and a new cancer drug toxicity data set is strong.

Weaknesses: - It would help to have some more justification of the statement, "much of the statistical flexibility of the Potts model is also retained" (line 98). - Table 1 is hard to read. Authors should use bold text or something else to distinguish best performers in each column. - A specific example of the "richer correlation structure" found by POIS (line 246) would be more convincing.

Correctness: Yes.

Clarity: Yes, very well written.

Relation to Prior Work: Yes.

Reproducibility: Yes

Additional Feedback: "The POIS" sounds a bit funny as an acronym for the Potts-Ising Model. Why not "PIM"? UPDATE: This paper stands out to me. It is well-written and highly motivated.

[Author Response · NeurIPS 2020]

We thank all four reviewers for their time and feedback. We will implement major changes, including redoing the simulations with a new evaluation metric: **[R2] Not beating Ind Mult (and [R1] methodology).** This was a major concern for us too. The issue was the specific evaluation metric we used following Inouye et. al. [12]. Their idea was to aggregate all the pairwise MMD (i.e., MMD between bi-variate marginals) to form a histogram. The issue is that pairwise dependence might not be strong for many pairs in real data. However, the overall distribution is still far from a product of multinomials due to higher-order dependencies. Ideally, one should compute MMD between full joint distributions of the "learned model" and the "test data". This however is a single number and we liked the idea of aggregation which gives a more robust metric. It occurred to us that we can retain this nice feature, while measuring higher-order dependencies much better, by looking at all the $\binom{d}{d-2}$ marginals instead of $\binom{d}{2}$. That is, assuming $d = 10$, we plan to measure the maximum discrepancy of the two distributions on moments of the form $\mathbb{E}f(X_{i_1}, X_{i_2}, \ldots, X_{i_8})$ for all $f$ in the unit ball of the RKHS and all $\{i_1, \ldots, i_8\} \subset \{1, \ldots, 10\}$. We refer to this as *pair-complement* MMD. This is in contrast to our current approach of looking only at moments of the form $\mathbb{E}f(X_{i_1}, X_{i_2})$. If the RKHS is rich enough, functions of the form $f(X_{i_1}, X_{i_2}, \ldots, X_{i_8})$ already include those of the form $f(X_{i_1}, X_{i_2})$ by being constant in the extra arguments. Below we have provided some figures with this new metric which clearly shows that POIS and bootstrap (as well as Copula Mult) significantly beat Ind Mult. **[R2] Copula Mult.** Thanks. Indeed, this is a natural choice and will be added to the simulations. As the sample figures show, it generally performs very well, and POIS is quite competitive with it. **[R2] Poisson models.** We agree, though they are not totally unreasonable when $\lambda$ is small, since Poisson concentrates ($\approx \lambda + O(\sqrt{\lambda})$). We will elaborate more in the paper. **[R2] Inverse Spearman corr.** We understand your concern and try to find a better way to compare (including with the graphical structure of Copula Mult.) **[R2] Why Coord. Dec. beats Cond. Like.?** Not clear if this is the case, esp. in light of the new metric.

**[R1,R2,R4] Inference results rather than MMD; qualitative/intuitive comparisons; interpretation.** It is difficult to evaluate unsupervised approaches on inference results because of the lack of a ground truth, and since different models estimate different parameters. MMD provides an objective measure of how close the learned model is to the empirical distribution of the test data, and can be uniformly applied to all methods. That being said, we plan to provide more detailed qualitative comparisons based on the estimated correlations, and whether the results agree with domain knowledge. In fact, our original motivation for writing this paper was the interesting correlations predicted by POIS in the toxicity data, which intuitively made sense based on co-occurrence of symptoms. We will elaborate more in the revision. We will also add some simulated data and compare with the ground truth. **[R1] Scalability beyond $d = 50$.** We plan to investigate scalability more thoroughly in the revision. **[R1] Sampling parameter $m = 1000$.** We are sampling from the learned model where there is no limitation. We will clarify more in the paper. **[R1,R2] Notation.** Thanks for your suggestions. We will simplify and clarify the notation and technical arguments. **[R3] Single data set.** We are looking at two different types of toxicity data sets (PRO and toxicity) as well as multiple rating and count data sets. We will try to expand more. **[R3] Class Correlation.** Our point here is that lumping together the correlations among nonzero levels/classes is a good approximation in some applications. It may not be as severe as it may seem. We agree that one can add more complexity (at the expense of interpretation and possible over-fitting). **[R4]** PIM = Probabilistic Index Model; other suggestion? **[R1,R2,R3,R4]** We will incorporate as much of the other suggestions as possible.



[Meta-Review · NeurIPS 2020]

Four expert reviewers placed this paper near the borderline. All reviewers recognized that the basic idea to define a model that strikes a middle ground between the Potts model and Ising model was simple and elegant, and that the results were technically correct. R1 and R4 especially liked the simplicity and elegance, and all reviewers were generally very positive about the idea being motivated by a real-world application. Overall, the reviewers were not surprised by the technical results given the similarity to existing models, but acknowledged their novelty. There were a number of questions/concerns/suggestions, especially relating to baselines, evaluation metic, and diversity of data sets in the experiment as well as dense and non-standard notation. The authors provided new experimental results, proposed a new evaluation metric, and compared to the Copula Multinomial baseline suggested by R2 in the rebuttal; these improvements convinced two reviewers to raise their scores. While some lesser questions/concerns remain (e.g. R3 is surprised/skeptical that an apparently impoverished model will work in a broad range of settings), all reviewers recognized positive aspects of the paper and two were enthusiastic. The authors are strongly encouraged to take the very thoughtful reviewer comments into account while revising the paper, especially relating to notation.